# The use of respiratory rate-oxygenation index to predict failure of high-flow nasal cannula in patients with coronavirus disease 2019-associated acute respiratory distress syndrome: A retrospective study

**Sujaree Poopipatpab**[1], **Pruchwilai Nuchpramool**[1], **Piyarat Phairatwet**[2], **Todspol Lertwattanachai**[3], **Konlawij Trongtrakul**[4]*

1 Department of Anesthesiology, Faculty of Medicine Vajira Hospital, Navamindradhiraj University, Bangkok, Thailand, 2 Department of Internal Medicine, Faculty of Medicine Vajira Hospital, Navamindradhiraj University, Bangkok, Thailand, 3 Department of Pharmacology, Faculty of Medicine Vajira Hospital, Navamindradhiraj University, Bangkok, Thailand, 4 Department of Internal Medicine, Faculty of Medicine, Chiang Mai University, Chiang Mai, Thailand

* konlawij@live.com

**Data Availability Statement:** We added the dataset as the Supporting information.

## Abstract

Patients with mild-to-moderate coronavirus disease 2019 (COVID-19)-associated acute respiratory distress syndrome (ARDS) can be treated with a high-flow nasal cannula (HFNC). The use of the respiratory rate-oxygenation (ROX) index, calculated as the ratio of oxygen saturation ($SpO_2$)/fractional oxygen ($FiO_2$) to respiratory rate, in the first few hours after HFNC initiation can help identify patients who fail HFNC therapy later. However, few studies have documented the use of the ROX index during the period of HFNC therapy. Therefore, we aimed to demonstrate the diagnostic performance of the ROX index when calculated throughout the HFNC therapy period and to determine the best cut-off point for predicting HFNC failure. We conducted a retrospective study of patients with COVID-19-associated ARDS who commenced HFNC at the Faculty of Medicine Vajira Hospital, Navamindradhiraj University, Thailand, between April 1 and August 30, 2021. We calculated the ROX index every 4 h throughout the HFNC therapy period and defined HFNC failure as a subsequent endotracheal tube intubation. The performance of the ROX index was analyzed using the area under the receiver operating characteristic curve (AUC). We applied the ROX index $\leq$ 4.88 to predict HFNC failure and obtained a new ROX cut-off point using Youden's method. In total, 212 patients with COVID-19 treated with HFNC were included in the study. Of these, 81 patients (38.2%) experienced HFNC failure. The ROX index $\leq$ 4.88 demonstrated a reasonable performance in predicting HFNC failure (AUC, 0.77; 95% confidence interval [CI], 0.72–0.83; p<0.001). However, compared with the original cut-off point of $\leq$ 4.88, the new ROX index cut-off point of $\leq$ 5.84 delivered optimal performance (AUC, 0.84; 95% CI, 0.79–0.88; p<0.001), with a significantly better discriminative ability (p = 0.007). In conclusion, a ROX index $\leq$ 5.84

**Funding:** This study was supported by the Navamindradhiraj University Research Fund (grant no. COA 149/64). The funders had no role in the study design, data collection, data analysis, decision to publish, and preparation of the manuscript.

**Competing interests:** The authors have declared that no competing interests exist.

was found to be optimal for predicting HFNC failure in patients with COVID-19-associated ARDS.

## Introduction

The novel coronavirus disease 2019 (COVID-19) has been rapidly spreading worldwide as a result of the severe acute respiratory syndrome coronavirus 2 infection [1]. Approximately 20% of patients with COVID-19 progress to critical conditions owing to a more severe form of acute respiratory distress syndrome (ARDS) [2, 3]. In the initial stages of the pandemic, over 70% of hospitalized patients with severe COVID-19 pneumonia were intubated and usually commenced on invasive mechanical ventilation (IMV) [4–6].

A high-flow nasal cannula (HFNC) can be utilized as early respiratory support for mild-to-moderate COVID-19-associated ARDS [7]. Although it was formerly regarded as an aerosol-generating procedure, several studies have proven that HFNC is safe and feasible and does not result in severe acute respiratory syndrome coronavirus 2 transmissions [8–13]. According to the most recent research available, the respiratory rate-oxygenation (ROX) index has been commonly used to predict which patients will be unsuccessfully treated with HFNC therapy [14–17]. This index is calculated as the ratio of oxygen saturation ($SpO_2$)/fractional inspired oxygen ($FiO_2$) to the respiratory rate (RR). For non-COVID-19 patients with acute hypoxemia respiratory failure (AHRF), an ROX index value greater than 4.88 at 12 h after HFNC treatment was associated with a lower risk of the endotracheal tube (ET) intubation [18, 19].

Several studies have also reported the use of the ROX index for predicting HFNC failure in patients with COVID-19 [15–17, 20]. The ROX index was calculated at various time points, such as 2 h, 6 h, and 12 h after HFNC initiation, with different cut-off points for predicting HFNC failure ranging from 2.70 to 5.99 [15–17, 20]. However, in clinical practice, uncertainty remains regarding whether the ROX index can predict HFNC failure in patients with COVID-19 during the HFNC treatment period. Therefore, this study aimed to examine the best ROX cut-off point for determining HFNC failure throughout the HFNC treatment period in patients with COVID-19-associated ARDS.

## Materials and methods

### Study design and participants

This study was conducted retrospectively on patients with confirmed COVID-19 who were admitted to the intensive care unit (ICU) and cohort ward of the Faculty of Medicine Vajira Hospital, Navamindradhiraj University, Bangkok, Thailand, between April 1 and August 30, 2021. This study was approved by the Vajira Institutional Review Board (COA number 149/2564) on July 19, 2021 and was performed in accordance with the Declaration of Helsinki as a statement of ethical principles for medical research involving human subjects. The requirement for informed consent was waived owing to minimal risk, and the data were extracted and analyzed anonymously.

Data were obtained from the medical records of patients with mild-to-moderate COVID-19-associated ARDS who had received HFNC therapy. The inclusion criteria comprised patients aged ≥ 18 years who had a confirmed diagnosis of COVID-19 using a positive reverse transcription polymerase chain reaction, bilateral infiltration on chest radiography, and underwent HFNC therapy during hospitalization. We excluded patients in whom IMV was initiated

before commencing HFNC therapy, those who signed a do-not-resuscitate advance medical directive, and those who were transferred to another hospital during the HFNC therapy.

HFNC therapy was initiated using the AIRVO-2 Nasal High Flow System (Fisher and Paykel Healthcare Ltd., Auckland, New Zealand), with an initial flow of 40–60 L/min, the temperature of 34°C-37°C, and $FiO_2$ of 60%. The physician titrated $FiO_2$ to a target $SpO_2$ of more than 92% and adjusted the flow rate according to the patient's comfort or maximum tolerance.

Patients who were unable to maintain an $SpO_2 > 88\%$ when the $FiO_2$ was maximum or did not achieve an RR decrease to $< 35$ breaths/min were transferred to the ICU for close monitoring, provided that the patient was admitted to the cohort ward. Patients deemed to have HFNC treatment failure received intubation at the discretion of the attending physician. Generally, the indications for ET intubation are respiratory distress, severe metabolic acidosis, altered mental status, and cardiac arrest.

### Data collection and definitions

We collected data on patient demographics, including age, sex, and body mass index as well as pre-existing comorbidities, including diabetes mellitus, hypertension, dyslipidemia, cerebrovascular disease, and chronic kidney disease. During the initial phase of HFNC therapy, we also collected data on patients' vital signs, including body temperature, heart rate, mean arterial pressure, RR, $SpO_2$, $SpO_2/FiO_2$, and severity score, as measured by the Sequential Organ Failure Assessment (SOFA) score, and basic laboratory investigations, including complete blood count, serum creatinine, D-dimer, and C-reactive protein levels. Additionally, we extracted the $SpO_2$, $FiO_2$, and RR every 4 h during HFNC therapy from the initiation to the end of HFNC treatment. We calculated and utilized all of the ROX index values during the entire HFNC treatment period to determine HFNC failure. The reason for HFNC termination can be either HFNC success or failure. For a patient who required ET intubation with an IMV, we recorded this case as HFNC failure. The total number of HFNC days was defined as the duration from the start of HFNC therapy until the end of HFNC treatment. Other outcomes were also evaluated, including ICU length of stay (ICU-LOS), hospital length of stay (Hosp-LOS), 28-day mortality rate, and hospital mortality rate.

### Sample size

The sample sizes were estimated parametrically based on the variance of the area under the receiver operating characteristic curve (AUC) and its marginal errors with a 95% confidence level for binormal assumption using the Hanley and McNeil formula [21]. Given a significance level of 0.05, and a power of the test at 80%, at least 211 cases were required for the study with an AUC of 0.74 and 95% confidence interval (CI) of 0.64–0.84 [18].

### Statistical analysis

Continuous variables were expressed as median and interquartile range (IQR). Categorical variables were expressed as frequencies and percentages. We used the Mann–Whitney U test to compare continuous variables and Fisher's exact test for categorical variables. All of the ROX index values during the period of HFNC therapy were used in our study. We assessed the performance of the ROX index in determining HFNC failure using receiver operating characteristic curves. We then calculated the area under the receiver operating characteristic curve (AUC) to demonstrate how the ROX index could determine HFNC failure. We defined the ROX index of $\leq 4.88$ as a reference cut-off point in determining HFNC failure across the period of HFNC therapy, according to the study by Roca et al. (2019) [19].

Additionally, we obtained the best ROX index cut-off point from our data using the Youden index (sensitivity + specificity -1) and compared its diagnostic performance (sensitivity, specificity, positive predictive value, negative predictive value, and AUC) with the previous cut-off point of 4.88 [19]. Therefore, a new purpose cut-off point could be performed equally at each 4 hourly time-point across the period of HFNC therapy. Furthermore, we compared the diagnostic performance of the ROX index with that of other respiratory variables, including RR, $SpO_2$, and $SpO_2/FiO_2$. A two-sided $p < 0.05$ was considered statistically significant. Statistical analyses were performed using STATA, version 16.0 (STATA Inc., College Station, TX, USA).

## Results

### Patient baseline characteristics

During the study period, 311 COVID-19 patients with AHRF who were treated with HFNC were admitted to our ICU and cohort wards. Ninety-nine patients were excluded for the following reasons: 66 patients with a do-not-resuscitate advance directive, 30 patients with ET intubation before using HFNC, and 3 patients who were transferred to other hospitals during the HFNC therapy (Fig 1). Accordingly, we gathered information from 212 patients; 81 of

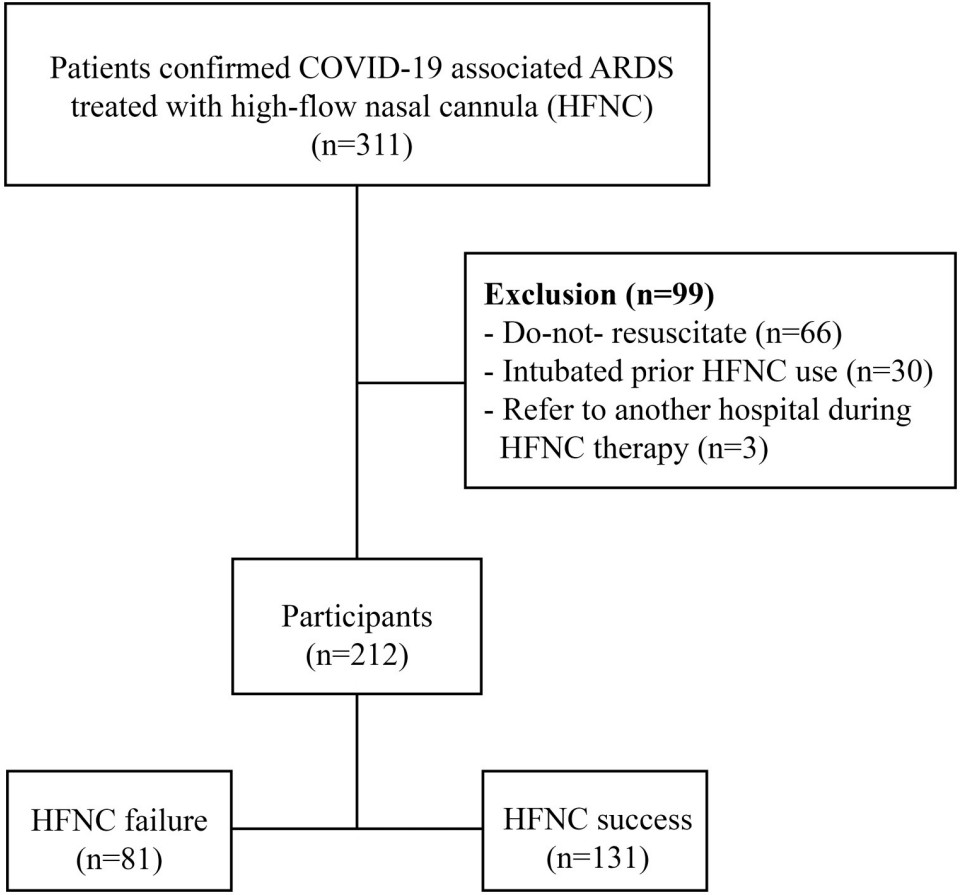

**Fig 1. Study flow diagram.** COVID-19, coronavirus disease 2019; ARDS, acute respiratory distress syndrome; HFNC, high-flow nasal cannula.

**Table 1. Baseline characteristics of patients with COVID-19 associated ARDS who commenced HFNC therapy.**

| Variables | All patients (n = 212) | HFNC failure (n = 81) | HFNC success (n = 131) | p value |
|---|---|---|---|---|
| *Demographic characteristics* | | | | |
| Age (years) | 62 (53–72) | 60 (53–72) | 63 (53–71) | 0.79 |
| Male, n (%) | 107 (50.5) | 45 (55.6) | 62 (47.3) | 0.25 |
| Body mass index, kg/m$^2$ | 26.7 (23.7–31.2) | 26.7 (23.7–31.3) | 26.7 (23.9–30.9) | 0.47 |
| *Pre-existing comorbidities, n (%)* | | | | |
| Diabetes mellitus | 98 (46.2) | 42 (51.9) | 56 (42.8) | 0.20 |
| Hypertension | 131 (61.8) | 50 (61.7) | 81 (61.8) | 0.99 |
| Dyslipidemia | 95 (44.8) | 39 (48.2) | 56 (42.8) | 0.44 |
| Cerebrovascular diseases | 21 (9.9) | 8 (9.9) | 13 (9.9) | 0.99 |
| Chronic kidney disease | 38 (17.9) | 19 (23.5) | 19 (14.5) | 0.10 |
| Symptoms to HFNC, days* | 7 (5–10) | 7 (4.5–9) | 8 (5–10) | 0.23 |
| *Vital signs and severity score at HFNC initiation* | | | | |
| Body temperature, ˚C | 36.9 (36.5–37.6) | 36.9 (36.4–37.9) | 36.9 (36.5–37.4) | 0.56 |
| Heart rate, beats/min | 80 (67–91) | 76 (66–95) | 81 (69–90) | 0.96 |
| Mean arterial pressure, mmHg | 93 (84–102) | 93 (82–102) | 92 (85–102) | 0.79 |
| Respiratory rate, breaths/min | 22 (20–24) | 22 (20–26) | 20 (20–24) | 0.006 |
| Pulse oxygen saturation, % | 96 (94–99) | 95 (93–97) | 97 (95–99) | 0.001 |
| SpO$_2$/FiO$_2$ ratio | 162 (157–208) | 160 (152–167) | 165 (158–238) | 0.001 |
| SOFA at HFNC initiation | 3 (2–3) | 3 (2–3) | 2 (2–3) | 0.22 |
| *Investigation at HFNC initiation* | | | | |
| Hemoglobin, g/dL | 12.8 (11.5–13.9) | 12.8 (11.5–14.2) | 12.9 (11.5–13.8) | 0.78 |
| White blood cell, 10$^3$ cells/mm$^3$ | 7.89 (5.57–11.62) | 7.93 (5.31–11.98) | 7.80 (5.82–11.19) | 0.48 |
| Absolute lymphocyte count, 10$^3$ cells/mm$^3$ | 0.80 (0.63–1.17) | 0.85 (0.62–1.13) | 0.79 (0.65–1.18) | 0.91 |
| Platelet count, 10$^3$ cells/mm$^3$ | 230 (183–303) | 208 (172–282) | 248 (191–332) | 0.01 |
| Creatinine, mg/dL | 0.88 (0.68–1.17) | 0.91 (0.79–1.19) | 0.83 (0.66–1.14) | 0.08 |
| D-dimer, ng/mL | 0.90 (0.49–2.03) | 0.86 (0.48–2.01) | 0.91 (0.49–2.04) | 0.85 |
| C-reactive protein, mg/L | 71.4 (37.3–109.0) | 65.7 (35.8–114.5) | 74.9 (39.5–109.0) | 0.77 |
| *Outcomes* | | | | |
| Length of HFNC use, days | 4 (2–7) | 2 (2–6) | 5 (3–7) | <0.001 |
| ICU admission, n (%) | 114 (53.8) | 78 (96.3) | 36 (27.5) | <0.001 |
| Length of ICU stay, days | 12 (6–20) | 14 (10–22) | 6 (5–10) | <0.001 |
| Length of hospital stay, days | 16 (10–24) | 24 (16–32) | 13 (9–17) | <0.001 |
| 28-day mortality, n (%) | 40 (18.9%) | 35 (43.2%) | 5 (3.8%) | <0.001 |
| Hospital mortality, n (%) | 53 (25.0%) | 47 (58.0%) | 6 (4.6%) | <0.001 |

Continuous data are presented as median (interquartile range) values.

ICU, intensive care unit; HFNC, high-flow nasal cannula; SOFA, Sequential Organ Failure Assessment score

these patients (38.2%) experienced HFNC failure. We found that most patients with HFNC failure were intubated within 2 days at a rate of 59.3% (48/81 cases), and almost 81.4% of patients (66/81 cases) received intubation within 7 days after the initiation of HFNC treatment (S1 Fig and S1 Table). None of the patients included in this study received noninvasive ventilation therapy in accordance with the first updated guidelines of the Surviving Sepsis Campaign on the management of critically ill adults with COVID-19 [13].

When comparing baseline characteristics between patients with HFNC failure and those with HFNC success, there were no significant differences in age, sex, body mass index, pre-existing comorbidities, number of days from symptoms to HFNC initiation, vital signs, and

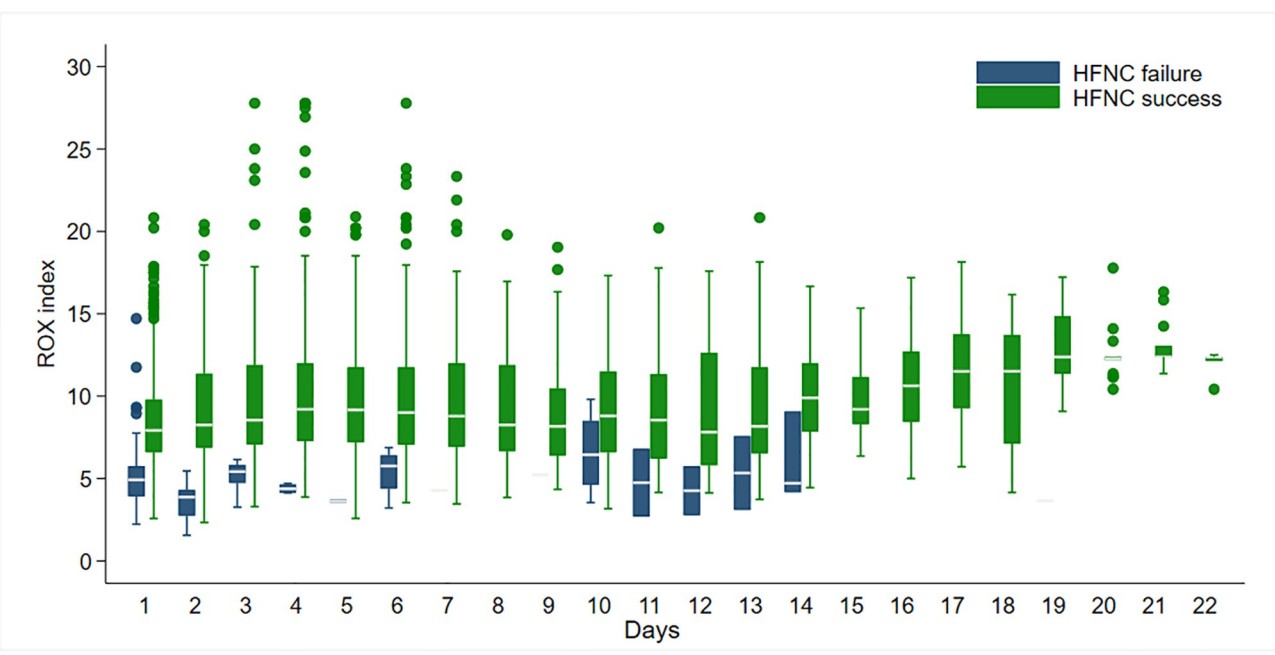

**Fig 2. Daily ROX index values between patients in the HFNC failure group and those in the HFNC success group.** HFNC, high-flow nasal cannula; ROX, respiratory rate-oxygenation.

laboratory investigations (Table 1). There were several exceptions, including higher RR, lower $SpO_2$, lower $SpO_2/FiO_2$, and lower platelet count in the HFNC failure group than in the HFNC success group (Table 1).

## Values of the ROX index during the study period

Fig 2 displays the daily ROX index comparison between patients in the HFNC failure and success groups in a longitudinal timeframe. Notably, we found that the ROX index values in the HFNC failure group for the initial 7 days ranged between 3.64 and 5.76. However, the ROX index values in the HFNC success group ranged from 7.92 to 9.20 (S1 Table).

Accordingly, during the initial 6 days, the ROX index in the HFNC failure group was significantly lower than that in the HFNC success group (all $p<0.05$; S1 Table). However, the ROX index did not differ significantly on day-7 ($p = 0.09$). This might be explained by only having one patient intubated on day-7.

## Outcomes between HFNC failure and success

The duration of HFNC use was shorter among patients with HFNC failure than among those with HFNC success (2 [IQR, 2–6] days vs. 5 [IQR, 3–7] days, $p<0.001$) (Table 1). In contrast, we found significantly longer ICU-LOS and Hosp-LOS in the HFNC failure group than in the HFNC success group, which were 14 (IQR, 10–22) days vs. 6 (IQR, 5–10) days, $p<0.001$ and 24 (IQR, 16–32) days vs. 13 (IQR, 9–17) days, $p<0.001$, respectively (Table 1). Furthermore, we also found significantly greater 28-day mortality and hospital mortality in the HFNC failure group than in the HFNC success group (43.2% vs. 3.8%, $p<0.001$ and 58.0% vs. 4.6%, $p<0.001$, respectively; Table 1).

## Accuracy of the ROX index in predicting HFNC failure

We found a strong diagnostic performance of the ROX index in its ability to predict HFNC failure, with an AUC of 0.89 and 95% CI of 0.85–0.93 (Fig 3). When an ROX index cut-off point of ≤ 4.88 was used to identify HFNC failure, the index still retained good discrimination ability (AUC, 0.78; 95% CI, 0.72–0.83; p<0.001; Table 2 and S2 Fig). Similarly, we found a more suitable ROX cut-off point using the Youden index at ≤ 5.84 in determining HFNC failure (AUC, 0.84; 95% CI, 0.79–0.88; p<0.001; Table 2 and S2 Fig). There was a significantly better discriminating ability at a cut-off point of 5.84 vs. 4.88, p = 0.007 (S2 Fig). Other diagnostic performance indicators, including sensitivity, specificity, positive predictive value, and negative predictive value, are shown in Table 2. Indeed, greater sensitivity was found at the new cut-off point (80.2% vs. 59.3%) than at the ROX cut-off point of 4.88. Hence, when applying an ROX index of ≤ 5.84 rather than 4.88, more cases would be correctly classified as HFNC failure with an earlier time determination. At a higher threshold, the ROX index could increase the level of confidence in decision-making when it comes to allocating patients, be it more intensive monitoring or more aggressive measures to be undertaken.

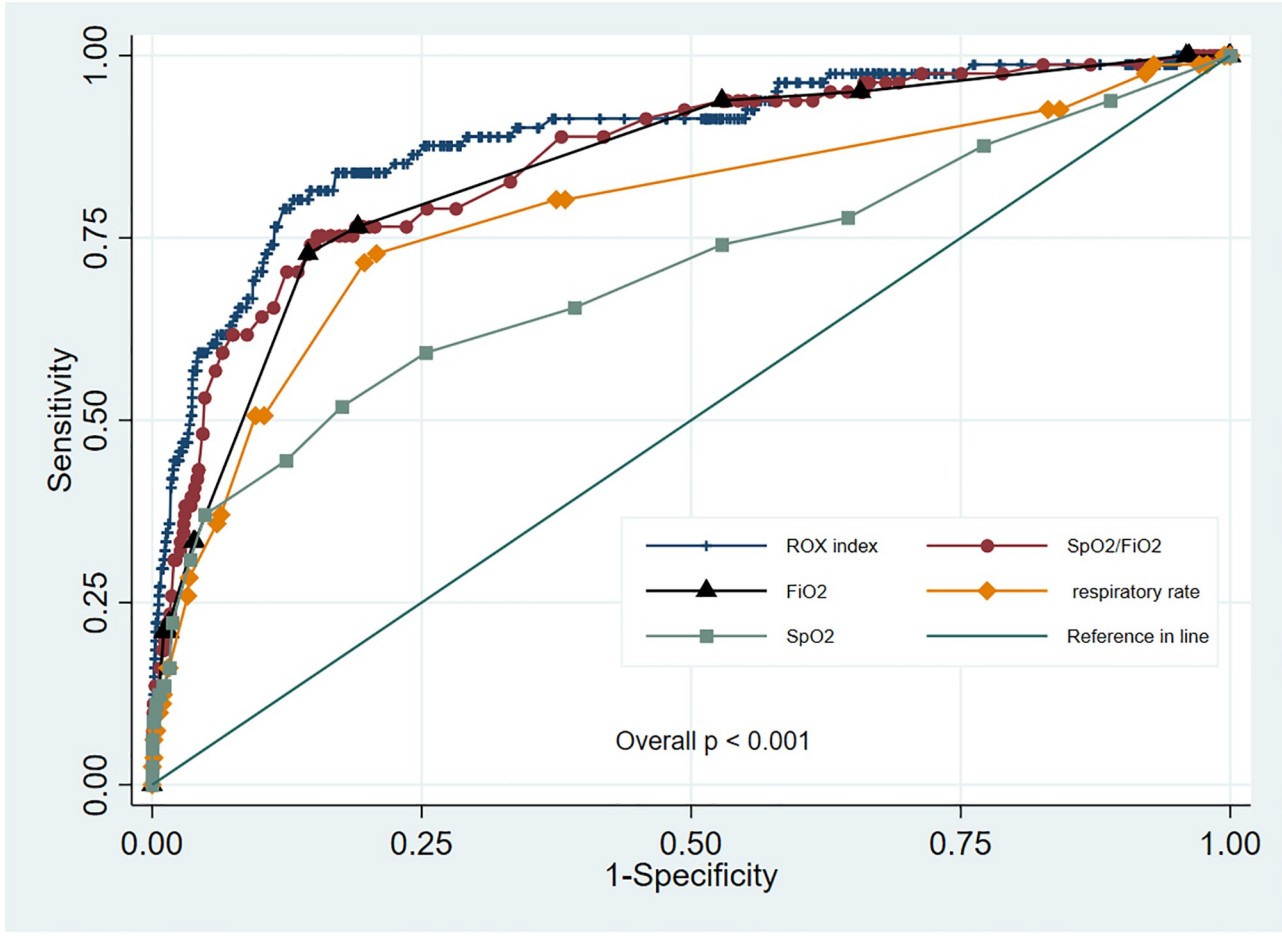

**Fig 3. The discriminative ability of ROX index, SpO$_2$/FiO$_2$, FiO$_2$, RR, and SpO$_2$ in determining HFNC failure.** HFNC, high-flow nasal cannula; ROX, respiratory rate-oxygenation; RR, respiratory rate.

**Table 2. Diagnostic performance of ROX index cut-offs in determining HFNC failure.**

| ROX cut-off | AUC (95% CI) | p value | Sensitivity (95% CI) | Specificity (95% CI) | PPV (95% CI) | NPV (95% CI) |
|---|---|---|---|---|---|---|
| ≤ 4.88 | 0.77 (0.72–0.83) | <0.001 | 59.3% (47.8%-70.1%) | 95.6% (95.1%-96.2%) | 16.6% (12.5%-21.3%) | 99.4% (99.1%-99.6%) |
| ≤ 5.84 | 0.84 (0.79–0.88) | <0.001 | 80.2% (69.9%-88.3%) | 86.9% (86.0%-87.8%) | 8.2% (6.4%-10.3%) | 99.7% (99.5%-99.8%) |

HFNC, high-flow nasal cannula; ROX, respiratory rate-oxygenation; AUC, area under the receiver operating characteristic curve; PPV, positive predictive value; NPV, negative predictive value

**Table 3. Diagnostic performance of the ROX index compared with that of other respiratory parameters in determining HFNC failure.**

| Parameters | AUC | 95% CI | p value | Overall p value | Pairwise comparison* |
|---|---|---|---|---|---|
| ROX index | 0.89 | 0.85–0.93 | <0.001 | <0.001 | Ref |
| SpO2/FiO2 | 0.86 | 0.81–0.90 | <0.001 | - | 0.03 |
| FiO2 | 0.84 | 0.80–0.89 | <0.001 | - | 0.001 |
| RR | 0.78 | 0.72–0.84 | <0.001 | - | <0.001 |
| SpO2 | 0.70 | 0.63–0.77 | <0.001 | - | <0.001 |

*Adjusted using the Bonferroni method.

HFNC, high-flow nasal cannula; ROX, respiratory rate-oxygenation; AUC, area under the receiver operating characteristic; $FiO_2$, fractional inspired oxygen; $SpO_2$, oxygen saturation; $SpO_2/FiO_2$, ratio of oxygen saturation/fractional inspired oxygen; RR, respiratory rate

## Comparing the ROX index with other variables in predicting HFNC failure

The ROX index and other respiratory parameters, including RR, $SpO_2$, and $SpO_2/FiO_2$ ratio every 4 h, are summarized in S1 Table. In discriminating HFNC failure from HFNC success in patients with COVID-19-associated ARDS, the ROX index performed significantly better than the other parameters, including the $SpO_2/FiO_2$ ratio, $FiO_2$, RR, and $SpO_2$, with AUC values of 0.89 vs. 0.86, 0.84, 0.78, and 0.70, respectively, with an overall p<0.001 (Fig 3). There were also significantly better AUC values from the ROX index than from the other parameters after pairwise comparison (all p<0.05) (Table 3).

## Discussion

HFNC therapy is a useful tool for treating patients with mild-to-moderate ARDS [16, 22]. Several patients were safe from the need for ET intubation and demonstrated an improved survival rate [23]. The original study by Roca et al. demonstrated the diagnostic performance of the ROX index in AHRF patients with non-COVID-19 pneumonia [18]. The ROX index of ≤ 4.88 at 12 h after the initiation of HFNC could accurately identify patients with HFNC failure (AUC, 0.74; 95% CI, 0.64–0.84) [18]. Additionally, the same author's group reported a suitable use of the ROX index at 2 h, 6 h, and 12 h after initiation of HFNC treatment, with levels of < 2.85, <3.47, and < 3.85, respectively, in terms of HFNC failure prediction [19]. The AUCs in the study ranged from 0.68 to 0.76 [19].

For COVID-19-associated ARDS, the *Surviving Sepsis Campaign guidelines* suggest using HFNC over conventional oxygen therapy or non-invasive positive pressure ventilation in mild-to-moderate cases [13]. Nonetheless, close monitoring and more aggressive management with ET intubation with IMV support are recommended when worsening respiratory symptoms occur [13]. However, the decision of whether or not patients with COVID-19-associated ARDS will fail from HFNC treatment remains challenging, as HFNC may cause a delay in ET

intubation and increase the mortality rate. The ROX index may be an option that could be utilized as a prognostic marker to predict HFNC failure.

We confirmed that the use of the ROX index provided robust evidence to support triaging of HFNC failure in patients with COVID-19-associated ARDS. In our data, 81 of 212 patients experienced HFNC failure (38.2%) during the study period of HFNC therapy. The AUC of the ROX index was 0.89 (95% CI, 0.85–0.93) in discriminating patients with HFNC failure from those with HFNC success during the study period of HFNC therapy (22 days in total). Although the ROX cut-off point was from the study where AHRF patients were not infected with COVID-19, the value of the ROX index of ≤ 4.88 could acceptably discriminate our COVID-19 patients with HFNC failure (AUC, 0.78; 95% CI, 0.72–0.83). Indeed, we found that the ROX index cut-off value of ≤ 5.84 was more optimum (AUC, 0.84; 95% CI, 0.79–0.88) and significantly better than that of 4.88 (p = 0.007).

Prior studies have reported the use of the ROX index to identify patients with COVID-19-associated ARDS who showed the failure of HFNC therapy [14, 16, 17, 24, 25]. The values of the ROX index in these studies were measured at several specific time points within the first 24 h after the initiation of HFNC treatment [16, 17, 24, 25]. A wide range of ROX cut-off points and AUC have also been reported for predicting HFNC failure. For example, the ROX index cut-off point < 4.94 measured at 2 to 6 h, with an AUC of 0.71 [16]; the ROX index < 5.40 measured within the first 4 h, with an AUC of 0.75 (95% CI, 0.60–0.91) [24]; the ROX index < 5.99 measured at 12 h, with an AUC of 0.79 (95% CI, 0.69–0.89) [17]; and the ROX index < 8.54 measured at 4 h, with an AUC of 0.70 (95% CI, 60–0.80) [25]. Moreover, a meta-analysis of patients with COVID-19 showed that a cut-off value of the ROX index above 5.00 provided more discriminative accuracy than that at 5.00 or below in predicting HFNC failure (p = 0.002) [26].

The above-mentioned evidence supports that an ROX cut-off point of < 5.84 is more suitable for patients with COVID-19 than an ROX cut-off point of < 4.88 in terms of HFNC failure detection. In addition, patients with COVID-19 may experience happy hypoxia [27]. A lower RR in AHRF patients with COVID-19 than in non-COVID-19 patients or a lower RR following HFNC therapy may be the reasons for the lowering of the denominator, RR, for the ROX index calculation.

Furthermore, recent studies have revealed that other variables such as $SpO_2/FiO_2$ [25] or RR [28] might be a better option for predicting HFNC failure than the use of the ROX index. However, we found that the ROX index had the most discriminative ability to predict HFNC failure compared with other respiratory parameters during the study period of HFNC treatment.

This study revealed that the ROX index at a value of ≤ 5.84 could be utilized to predict HFNC failure throughout the HFNC therapy period in patients with COVID-19-associated ARDS. However, the conventional value of the ROX index cut-off point of ≤ 4.88 might be less justified for triggering more aggressive management. Additionally, we confirmed that the ROX index was more accurate than other respiratory parameters.

Commonly, HFNC is continued long enough until patients' signs and symptoms of respiratory failure improve or until they clinically worsen, requiring more aggressive intervention. The decision for aggressive management based on the use of a single-initial ROX index after HFNC therapy in determining HFNC failure might not be valid in daily clinical practice, especially in patients who underwent a long period of HFNC therapy. A single-initial value of the ROX index may promote unnecessary proactive management in some patients. A repeated evaluation of the ROX index routinely every 4–6 h during the period of HFNC therapy as well as vital sign assessment might be an option for more appropriate allocation of patients at high risk of HFNC failure than of those with low risk.

However, this study had some limitations. First, this study was performed retrospectively and was reported from a single center. Therefore, further multicenter prospective studies are required to prove the benefit of an ROX index of $\leq 5.84$ in triaging HFNC failure. Second, this study analyzed the overall outcome of ROX index values in determining HFNC failure using conventional AUC estimation. AUC summarizes the capacity of the test to discriminate disease from non-disease across all possible levels of positivity into a single statistic. One might argue that using the conventional AUC estimation for a longitudinal measurement or time-dependent variable, that is, the ROX index value every 4 h during the HFNC therapy period, might lead to a statistical fault. Unfortunately, until recently, there has been a paucity of suitable methods or well-recognized statistics to handle repeated measures of time-dependent variables [29]. Third, patients with HFNC failure might have experienced respiratory distress rather than COVID-19 pneumonia progression, particularly those with late HFNC failure. However, almost 80% of our patients were intubated within 7 days after the initiation of HFNC therapy, remaining within the peak of inflammation or cytokine storm phase of COVID-19-associated ARDS. Finally, a high rate of intubation (38.2%) as well as high 28-day mortality and hospital mortality rates (18.9% and 25.0%, respectively) in our setting might indicate suboptimal monitoring and delayed intubation; however, this could have been due to patients having to be admitted to cohort wards instead of the ICU during the study period. Moreover, several studies have reported rates of HFNC failure that are similar to that in our study, ranging from 38.1% to 66.1% in patients admitted to the ICU [20, 24, 28]. Additionally, Chandel et al. recently conducted a retrospective analysis in a mixed-population of ICU and non-ICU COVID-19 patients and reported results similar to ours (ICU admission, intubation, mortality, and mortality with HFNC failure rates of 66.7%, 39.7%, 18.0%, and 45.4%, respectively) [30].

## Conclusion

In patients with COVID-19-associated ARDS, the use of the ROX index of $\leq 5.84$ during the period of HFNC therapy was the optimal cut-off point for determining those with HFNC failure. In addition, this new cut-off point of 5.84 was significantly better than the original value of 4.88. Additionally, we found that the ROX index was more accurate in revealing the HFNC failure than other respiratory variables.

## Supporting information

**S1 Fig. Number of daily HFNC failure in patients with COVID-19 associated ARDS.**
(TIFF)

**S2 Fig. Comparison AUC of ROX index cut-off point value 4.88 vs 5.84.**
(TIFF)

**S1 Table. Respiratory parameters within 7 days after the initiation of HFNC treatment.**
(DOCX)

**S1 File. Study protocol.**
(DOCX)

**S2 File. Study dataset.**
(XLSX)

## Acknowledgments

We would like to thank Miss Unyaparch Rungsidhavaspong and Miss Panithi Pongraweewan for data gathering and Mr. Anucha Kamsom for his assistance with statistical analysis. We sincerely thank Jonathan Stream (Ba.), an English Consultant, for English proofreading assistance.

## Author Contributions

**Conceptualization:** Sujaree Poopipatpab, Konlawij Trongtrakul.

**Data curation:** Sujaree Poopipatpab, Pruchwilai Nuchpramool, Piyarat Phairatwet, Todspol Lertwattanachai.

**Formal analysis:** Sujaree Poopipatpab, Konlawij Trongtrakul.

**Funding acquisition:** Sujaree Poopipatpab.

**Writing – original draft:** Sujaree Poopipatpab, Konlawij Trongtrakul.

**Writing – review & editing:** Konlawij Trongtrakul.

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
