## [Decision Letter · Decision Letter 0]

6 Apr 2023

PONE-D-23-04347The use of respiratory rate-oxygenation index to predict failure of high-flow nasal cannula in patients with coronavirus disease 2019-associated acute respiratory distress syndrome: A retrospective studyPLOS ONE

Dear Dr. Trongtrakul,

Thank you for submitting your manuscript to PLOS ONE. After careful consideration, we feel that it has merit but does not fully meet PLOS ONE’s publication criteria as it currently stands. Therefore, we invite you to submit a revised version of the manuscript that addresses the points raised during the review process.

We look forward to receiving your revised manuscript.

Kind regards,

Vijay Hadda, MD

Academic Editor

PLOS ONE

Additional Editor Comments:

Major comments:

1. Authors have stated that data related to the SpO2, FiO2, and RR (to calculate the ROX index) were extracted every 4 hourly during HFNC therapy from the initiation to the end of HFNC treatment. Thus there were multiple ROX value (6 values every day). However, authors have not mentioned which ROX value was used. Which 4 hourly ROX value was used to calculate failure prediction at each time point? Or the mean value of entire day ROX was used? Was the proposed cut-off perform equally at each 4 hrly time-point? Please clarify in the manuscript.

2. Among the selection criteria, authors have not mentioned about NIV use. Was there no patient who received NIV? Or NIV use was exclusion criteria? Please clarify

Reviewers' comments:

Reviewer's Responses to Questions

**Comments to the Author**

1. Is the manuscript technically sound, and do the data support the conclusions?

Reviewer #1: Yes

Reviewer #2: Partly

2. Has the statistical analysis been performed appropriately and rigorously? 

Reviewer #1: Yes

Reviewer #2: Yes

3. Have the authors made all data underlying the findings in their manuscript fully available?

Reviewer #1: Yes

Reviewer #2: Yes

4. Is the manuscript presented in an intelligible fashion and written in standard English?

Reviewer #1: Yes

Reviewer #2: Yes

5. Review Comments to the Author

Reviewer #1: The authors have presented the new ROX index in COVID 19. It has been a matter of debate whether the same ROX index is applicable to Non COVID and COVID ARDS. The manuscript is well written with detailed statistical analysis.

Reviewer #2: The authors seem to allude to using HFNO in sick patients on the wards which is not usual practice. They talk about maximum oxygen (which would imply close to 100% FiO2) and saturations of 88% as an indication for transfer to ICU! One would argue this would be an indication to consider intubation in a carefully monitored patient in ICU. The authors use unconventional indications of intubation and ventilation. There is no mention of PaO2/FiO2 ratio as an indication of intubation and ventilation. ‘Cardiac arrest’ is mentioned as an indication for intubation!

There is also no mention of NIV in the algorithm. Has HFNO been used in a center with no access to NIV? Is this being suggested as a replacement for NIV?

One could argue that the high rate of HFNO failure with resultant intubation and ventilation was because the patients were on the wards while having high oxygen requirements. The use of HFNO on relatively unmonitored wards with low health care worker inputs (supposedly) might have been a reason of delayed intubation and ventilation. The above factors could have influenced the ROX score calculated. The high overall mortality (25%) and the very high mortality in the patients intubated and ventilated (58%) would also indicate suboptimal monitored and delayed intubation and ventilation.

However, I appreciate the efforts to carry out an interesting study during the very difficult COVID times. I feel this study would also be a ‘proof of concept study’ for pother Viral Pneumonias. Hence a similar study, which is better planned should be carried out for this other category of patients.

6. PLOS authors have the option to publish the peer review history of their article (what does this mean?). If published, this will include your full peer review and any attached files.

Reviewer #1: No

Reviewer #2: No

---

## [Author Response · Author response to Decision Letter 0]

15 May 2023

Manuscript ID: PONE-D-23-04347

9 May, 2023

Emily Chenette

Editor-in-Chief

PLOS ONE 

Dear Editors and Reviewers:

I wish to re-submit the manuscript titled “The use of respiratory rate-oxygenation index to predict failure of high-flow nasal cannula in patients with coronavirus disease 2019-associated acute respiratory distress syndrome: a retrospective study”. 

We thank you and the reviewers for your thoughtful suggestions and insights, which have enhanced the quality of our research significantly. We have followed the recommendations line-by-line and have done our best to make certain the revision meets the journal’s requirements. For details on the modifications, please refer to the responses to the comments listed below. In case there are any other queries or concerns, please let us know, and we will be happy to address them.

Regards, 

Konlawij Trongtrakul

Faculty of Medicine, Chiang Mai University, Chiang Mai, Thailand.

 

Responses to Editor and Reviewers

Editor Comments to the Authors:

Response: Thank you for this information. We have now revised the manuscript and other files in accordance with the PLOS ONE style requirements. 

Response: Thank you for the information. This study was supported by the Navamindradhiraj University Research Fund (grant no. COA 149/64). The funders had no role in the study design, data collection, data analysis, decision to publish, and preparation of the manuscript.

Response: Thank you for the information. We have now uploaded the anonymized data set as a Supporting Information file. 

Response: Thank you for this information. This retrospective cohort study was performed in accordance with the standards stated in the Declaration of Helsinki and approved by the Vajira Institutional Review Board (COA number 149/2564). The requirement for informed consent was waived, as the information was gathered anonymously and posed minimal risk to the participants (lines 74-81).

Major comments:

1. Authors have stated that data related to the SpO2, FiO2, and RR (to calculate the ROX index) were extracted every 4 hourly during HFNC therapy from the initiation to the end of HFNC treatment. Thus there were multiple ROX value (6 values every day). However, authors have not mentioned which ROX value was used. Which 4 hourly ROX value was used to calculate failure prediction at each time point? Or the mean value of entire day ROX was used? Was the proposed cut-off perform equally at each 4 hourly time-point? Please clarify in the manuscript.

Response: We used the entire ROX values during the period of HFNC treatment for the prediction of HFNC failure (lines 110-111 & 131-132). The proposed ROX cut-off point in our study perform equally at each 4 hourly time-point (lines 136-139).

2. Among the selection criteria, authors have not mentioned about NIV use. Was there no patient who received NIV? Or NIV use was exclusion criteria? Please clarify

Response: According to the first update of the guidelines of the Surviving Sepsis Campaign on the management of adults with Coronavirus Disease 2019 in the ICU, it is suggested that HFNC be used instead of NIV for adults with COVID-19 and acute hypoxemic respiratory failure [1]. Therefore, we did not utilize NIV as the rescue therapy at our center at the time of data collection. We have now clarified this in the Results section of the revised manuscript (lines 153-155).

Reviewers' Comments to the Authors:

#Reviewer 1

The authors have presented the new ROX index in COVID 19. It has been a matter of debate whether the same ROX index is applicable to Non COVID and COVID ARDS. The manuscript is well written with detailed statistical analysis.

Response: Thank you very much for kind review of our manuscript.

#Reviewer 2

The authors seem to allude to using HFNO in sick patients on the wards which is not usual practice. They talk about maximum oxygen (which would imply close to 100% FiO2) and saturations of 88% as an indication for transfer to ICU! One would argue this would be an indication to consider intubation in a carefully monitored patient in ICU. The authors use unconventional indications of intubation and ventilation. There is no mention of PaO2/FiO2 ratio as an indication of intubation and ventilation. ‘Cardiac arrest’ is mentioned as an indication for intubation!

There is also no mention of NIV in the algorithm. Has HFNO been used in a center with no access to NIV? Is this being suggested as a replacement for NIV?

One could argue that the high rate of HFNO failure with resultant intubation and ventilation was because the patients were on the wards while having high oxygen requirements. The use of HFNO on relatively unmonitored wards with low health care worker inputs (supposedly) might have been a reason of delayed intubation and ventilation. The above factors could have influenced the ROX score calculated. The high overall mortality (25%) and the very high mortality in the patients intubated and ventilated (58%) would also indicate suboptimal monitored and delayed intubation and ventilation.

However, I appreciate the efforts to carry out an interesting study during the very difficult COVID times. I feel this study would also be a ‘proof of concept study’ for pother Viral Pneumonias. Hence a similar study, which is better planned should be carried out for this other category of patients.

Response: Thank you for your comments and for your detailed review of our work.

1. Although Thailand was ranked in the top 5 countries that had a good response to the COVID-19 pandemic by Johns Hopkins University [1], we were exposed to a large wave of COVID-19 cases during the study period, and at that time some of our COVID-19 patients were admitted to isolation wards instead of the ICU. This situation might have led to suboptimal care, which may be a limitation of the present study. Nonetheless, Table 1 presents the overall SpO2, SP ratio, and RR values (96% [94%-99%], 162 [157-208], and 22 [20-24] bpm, respectively). After recalculating, the overall FiO2 in our cohort should be around 0.59 [96/162] (0.48 [99/208] to 0.60 [94/157]). Therefore, the maximum oxygen support (FiO2 =100%) and SpO2 < 88% as the main indication for transferring the patients to the ICU would be a rare case.

2. In our cohort, limited access to the patients in the ward and to continuous arterial line monitoring limited the possibility of obtaining arterial blood gas (ABG) data. Therefore, practically, we could not perform ABG monitoring routinely to assess the PF ratio levels before intubation.

3. We did not use NIV as an option for respiratory support for our patients according to the guidelines from the Surviving Sepsis Campaign on the management of adult with Coronavirus Disease 2019 in the ICU, which suggest using HFNC over NIV for adults with COVID-19 and acute hypoxemic respiratory failure [2]. Additionally, the awareness of aerosol generation from the use of the NIV at that time impeded the use of this technique at our center. We have now clarified this in our manuscript in the Result section (lines 153-155).

4. It might be true that a high rate of HFNC failure (at almost 40%) in our study could be explained by the use of HFNC in the ward instead of ICU admission. However, several reports show that HFNC failure rate in patients who were admitted to the ICU in other studies was not very different from that in our study. The overall mortality rate in our study and in patients with HFNC failure in previous reports were also similar. Additionally, a recent retrospective study by Chandel et al. on a mixed-population non-ICU and ICU population has also reported fairly similar results as ours in terms of ICU admission (66.7% and 53.8%), intubation rate (39.7% and 38.2%), overall mortality rate (18.0% and 18.9%), and mortality rate with HFNC failure (45.4% and 43.2%). We have discussed this issue as a limitation of our study (lines 313-322).

Study, Total (n), Setting, Definition of HFNC failure, HFNC failure, Overall mortality, HFNC failure mortality

Hu et al. [3], 105, ICU, ET with HFNC Rx, 38.1%, -, -

Panadero et al. [4], 40, Intermediate RCU, ET within D-30, 52.5%, 22.5%, 42.8%

Vega et al. [5], 120, Outside ICU, ET with HFNC Rx, 29.2%, -, -

Blez et al. [6], 30, ICU, ET within D-7, 53.3%, -, -

Zucman et al. [7], 62, ICU, ET with HFNC Rx, 66.1%, -, -

Chandel et al. [3], 272, Mixed ICU and non-ICU, ET with HFNC Rx, 39.7%, 18.0%, 45.4%

Our study, 212, Mixed ICU and ward, ET with HFNC Rx, 38.2%, 18.9%, 43.2%

D-7, day-7; D-30, day-30; ET, endotracheal tube intubation; HFNC Rx, high-flow nasal cannula therapy; ICU, intensive care unit; RCU, respiratory care unit. 

References

1. Alhazzani W, Møller MH, Arabi YM, Loeb M, Gong MN, Fan E, et al. Surviving Sepsis Campaign: guidelines on the management of critically ill adults with Coronavirus Disease 2019 (COVID-19). Intensive Care Med. 2020;46(5):854-87.

2. Global Health Secruity Index Country for Thailand. 2021 [Access date 14 April 2023]. Available from: https://www.ghsindex.org/country/thailand/.

3. Hu M, Zhou Q, Zheng R, Li X, Ling J, Chen Y, et al. Application of high-flow nasal cannula in hypoxemic patients with COVID-19: a retrospective cohort study. BMC Pulm Med. 2020;20(1):324.

4. Panadero C, Abad-Fernández A, Rio-Ramirez MT, Acosta Gutierrez CM, Calderon-Alcala M, Lopez-Riolobos C, et al. High-flow nasal cannula for acute respiratory distress syndrome (ARDS) due to COVID-19. Multidiscip Respir Med. 2020;15(1):693.

5. Vega ML, Dongilli R, Olaizola G, Colaianni N, Sayat MC, Pisani L, et al. COVID-19 pneumonia and ROX index: time to set a new threshold for patients admitted outside the ICU. Pulmonology. 2022;28(1):13-7.

6. Blez D, Soulier A, Bonnet F, Gayat E, Garnier M. Monitoring of high-flow nasal cannula for SARS-CoV-2 severe pneumonia: less is more, better look at respiratory rate. Intensive Care Med. 2020;46(11):2094-5.

7. Zucman N, Mullaert J, Roux D, Roca O, Ricard JD. Prediction of outcome of nasal high flow use during COVID-19-related acute hypoxemic respiratory failure. Intensive Care Med. 2020;46(10):1924-6.

8. Chandel A, Patolia S, Brown AW, Collins AC, Sahjwani D, Khangoora V, et al. High-flow nasal cannula therapy in COVID-19: using the ROX index to predict success. Respir Care. 2021;66(6):909-19.

---

## [Decision Letter · Decision Letter 1]

6 Jun 2023

The use of respiratory rate-oxygenation index to predict failure of high-flow nasal cannula in patients with coronavirus disease 2019-associated acute respiratory distress syndrome: a retrospective study

PONE-D-23-04347R1

Dear Dr. Trongtrakul,

We’re pleased to inform you that your manuscript has been judged scientifically suitable for publication and will be formally accepted for publication once it meets all outstanding technical requirements.

Kind regards,

Vijay Hadda, MD

Academic Editor

PLOS ONE

**Comments to the Author**

1. If the authors have adequately addressed your comments raised in a previous round of review and you feel that this manuscript is now acceptable for publication, you may indicate that here to bypass the “Comments to the Author” section, enter your conflict of interest statement in the “Confidential to Editor” section, and submit your "Accept" recommendation.

Reviewer #1: All comments have been addressed

2. Is the manuscript technically sound, and do the data support the conclusions?

Reviewer #1: Yes

3. Has the statistical analysis been performed appropriately and rigorously? 

Reviewer #1: Yes

4. Have the authors made all data underlying the findings in their manuscript fully available?

Reviewer #1: (No Response)

5. Is the manuscript presented in an intelligible fashion and written in standard English?

Reviewer #1: Yes

6. Review Comments to the Author

Reviewer #1: The revised manuscript has responded to all queries. The language has been edited appropriately. The manuscript is clinically relevant

7. PLOS authors have the option to publish the peer review history of their article (what does this mean?). If published, this will include your full peer review and any attached files.

Reviewer #1: No

---

## [Editor Report · Acceptance letter]

12 Jun 2023

PONE-D-23-04347R1 

The use of respiratory rate-oxygenation index to predict failure of high-flow nasal cannula in patients with coronavirus disease 2019-associated acute respiratory distress syndrome: a retrospective study 

Dear Dr. Trongtrakul:

I'm pleased to inform you that your manuscript has been deemed suitable for publication in PLOS ONE. Congratulations! Your manuscript is now with our production department. 

Kind regards, 

on behalf of

Dr. Vijay Hadda 

Academic Editor

PLOS ONE